# Cellular Targets of HIV-1 Protease: Just the Tip of the Iceberg?

**DOI:** 10.3390/v15030712

**Published:** 2023-03-09

**Authors:** Matteo Centazzo, Lara Manganaro, Gualtiero Alvisi

**Affiliations:** 1Department of Molecular Medicine, University of Padova, 35121 Padova, Italy; 2INGM, Istituto Nazionale Genetica Molecolare “Romeo ed Enrica Invernizzi”, 20122 Milan, Italy; 3Department of Pharmacological and Biomolecular Sciences, University of Milan, 20133 Milan, Italy

**Keywords:** HIV-1 PR, host factors, host cell shut-off, protease, antiviral therapy, cell death, apoptosis

## Abstract

Human immunodeficiency virus 1 (HIV-1) viral protease (PR) is one of the most studied viral enzymes and a crucial antiviral target. Despite its well-characterized role in virion maturation, an increasing body of research is starting to focus on its ability to cleave host cell proteins. Such findings are apparently in contrast with the dogma of HIV-1 PR activity being restricted to the interior of nascent virions and suggest catalytic activity within the host cell environment. Given the limited amount of PR present in the virion at the time of infection, such events mainly occur during late viral gene expression, mediated by newly synthesized Gag-Pol polyprotein precursors, rather than before proviral integration. HIV-1 PR mainly targets proteins involved in three different processes: those involved in translation, those controlling cell survival, and restriction factors responsible for innate/intrinsic antiviral responses. Indeed, by cleaving host cell translation initiation factors, HIV-1 PR can impair cap-dependent translation, thus promoting IRES-mediated translation of late viral transcripts and viral production. By targeting several apoptotic factors, it modulates cell survival, thus promoting immune evasion and viral dissemination. Additionally, HIV-1 PR counteracts restriction factors incorporated in the virion that would otherwise interfere with nascent virus vitality. Thus, HIV-1 PR appears to modulate host cell function at different times and locations during its life cycle, thereby ensuring efficient viral persistency and propagation. However, we are far from having a complete picture of PR-mediated host cell modulation, which is emerging as a field that needs further investigation.

## 1. Introduction

Human immunodeficiency virus 1 (HIV-1)—the causative agent of acquired immunodeficiency syndrome (AIDS)—is one of the most infamous viruses known to man. Since its adaptation to humans at the beginning of the 20th century [1], more than 84 million people have been infected, resulting in one of the largest epidemics in human history [2]. Virus isolation and characterization, together with exponential growth in the emergence of AIDS cases, ignited the quest to discover effective antiviral treatments. The first drugs to be discovered were inhibitors of the reverse transcriptase; however, these alone were not able to control the infection completely due to the rapid selection of resistant strains and high toxicity [3,4,5]. Of paramount importance in the fight against AIDS was the discovery of new antivirals capable of inhibiting the viral protease (PR), an essential enzyme encoded in the HIV-1 genome that catalyzes the maturation of the nascent virion: a great target for therapeutic compound development. Approval of PR inhibitors enabled the implementation of combination therapy entailing the administration of multiple drugs against different viral targets, ultimately leading to better control of the infection and granting HIV-1 infected people with a life comparable to that of uninfected individuals [6,7,8].

HIV-1 is a lentivirus belonging to the family *Retroviridae*. Its virion has a spherical shape with a diameter of roughly 100 nm and is composed of a cone-shaped protein capsid surrounded by a lipoprotein-rich envelope [9]. The positive single-strand RNA viral genome encodes for the three main genes common to all retroviruses, *gag*, *pol*, and *env,* as well as different accessory genes and regulatory elements (Figure 1). Gag and Pol are synthetized from a 9 kb unspliced mRNA, while Env is generated from a single-spliced 4 kb mRNA [10]. These genes are all expressed as polyprotein precursors and processed into their mature products either by the viral protease (in the case of Gag and Gag-Pol) or by a cellular protease (in the case of Env) [11]. Gag is composed of the matrix protein (MA) that associates with the inner layer of the viral envelope and the capsid, the capsid protein (CA) that makes up the cone-shaped protein capsid, and finally by the nucleocapsid protein (NC) that is found in the core and is tightly associated with two copies of the viral genome [12]. The *pol* gene is expressed after a ribosomal frameshift event resulting in the expression of the Gag-Pol polyprotein. *pol* encodes for three viral enzymes: the aforementioned protease (PR), the reverse transcriptase (RT) that catalyzes the conversion of the single stranded positive RNA genome into a double stranded DNA molecule, and the integrase (IN) that is responsible for the integration of the retro-transcribed proviral DNA in the genome of the host cell [13]. Lastly, the *env* gene encodes for the glycoproteins that are embedded in the viral envelope, such as the gp160 glycoprotein that is subsequently processed by cellular enzymes into its mature components: gp120 (surface or SU) and gp41 (transmembrane or TM) [11]. Apart from the three main ORFs, HIV-1 encodes for several accessory proteins that are of paramount importance for the viral life cycle: most of them are expressed by single-spliced transcripts, whereas transactivator of transcription (Tat), negative factor (Nef), and regulator of expression of virion proteins (Rev) are expressed from double-spliced transcripts and are therefore the first viral proteins to be expressed in the infected cell. Indeed, Tat is fundamental for elongation of viral mRNA transcripts while Nef is responsible for potently enhancing infectivity by protecting infected cells from cytotoxic CD8-mediated killing, counteracting the restriction factor SERINC5, and altering the response to T-cell receptor (TCR) stimulation, thus increasing LTR-dependent transcription and viral replication [14,15,16]. In contrast, Rev is responsible for the export of single-spliced and unspliced viral transcripts from the nucleus to the cytoplasm, allowing temporal regulation of viral gene expression. Subsequently, the other viral accessory proteins can be expressed: viral infectivity factor (Vif), which counteracts the antiviral activity of APOBEC3 proteins; viral protein R (Vpr), which promotes viral replication and is involved in cell cycle G2 arrest, apoptosis, and depletion of regulatory T-cells; and viral protein U (Vpu), a transmembrane protein that promotes viral replication by downregulating CD4 and tetherin-BST2 [17,18,19].

The HIV-1 replication cycle (Figure 2) starts with binding and entry into the target cell, which is mediated by interaction of viral glycoprotein gp120 trimers with virus receptors (CD4) and co-receptors (CCR5 or CXCR4) on the cell surface. Such events lead to gp41 exposure and ultimately to fusion of the viral and cellular membranes, thus allowing the viral particle to enter the cell [20]. Upon entry and formation of the reverse transcription complex (RTC), the viral RNA genome is reverse transcribed into a double stranded DNA molecule by the RT [18]. At the end of the reverse transcription process, the RTC is replaced by the pre-integration complex (PIC), which mediates the transport and consequent import of the viral DNA into the nucleus where it is integrated in the host’s genome thanks to the activity of the viral IN [9]. Subsequently, the proviral genome is expressed by the cellular machinery. The first viral mRNAs exported from the nucleus and translated are those that undergo two splicing events, namely those encoding for Tat, Rev, and Nef (Figure 1). These proteins are necessary for initiation and regulation of the transcription process and production of other viral proteins. Proviral transcription starts from the U3 region of the 5′ LTR of the integrated genome and requires Tat to produce full length transcripts. Meanwhile, Rev is fundamental for the export of unspliced or single-spliced immature mRNAs from the nucleus, allowing all viral proteins to be expressed [21]. Env is synthetized in the ER where it is glycosylated and cleaved into its two components by cellular proteases, finally travelling through the Golgi apparatus to the plasma membrane. Gag and Gag-Pol are synthesized in the cytoplasm from full-length, unspliced viral mRNA; after synthesis, the MA domain undergoes myristoylation at the N-terminus, thereby allowing polyprotein targeting to the cell membrane [22]. Viral genomic RNA is recruited to sites of viral assembly by interacting with the NC portion of Gag via the packaging signal Ψ present on the viral genome. Once all the components of the nascent virions are concentrated in proximity to the cell membrane, the Gag polyprotein starts to multimerize, thus forming the immature structure of the budding virion. The formation of these Gag-Gag interactions initiates viral budding, coordinated by the p6 late (L) domain and the cellular endosomal sorting complex required for transport (ESCRT), thus allowing the formation and budding of the viral particle. Release of newly formed virions is inhibited by the cellular transmembrane factor tetherin-BST2, but this restriction is promptly averted by intervention by the viral transmembrane protein Vpu [21,23,24]. Finally, the action of the viral PR catalyzes maturation of the virion to infect other target cells. Interestingly, HIV-1 can infect target cells by both cell-free and cell-to-cell transmission through the formation of virological synapses, which are sites of cell-to-cell contact that direct viral transfer [25,26,27,28]. Notably, cell-to-cell transmission has been shown to be more efficient than cell-free transmission. Nevertheless, cell-to-cell transmission retains sensitivity to the majority of non-nucleoside-analog reverse transcriptase inhibitors (NNRTI) as well as protease and entry inhibitors [29,30].

### 1.1. HIV-1 PR Structure and Function

HIV-1 PR is a homodimeric aspartic protease composed of two monomers, each composed by 99 amino acids (Figure 3). Each monomer contains one α-helix and nine β-strands, four of which are antiparallel sheets and make up the highly-stable dimer interface, which in turn forms the active site [31]. The latter is characterized by a hydrophobic core and contains the Asp-Thr-Gly catalytic triad - which is common among different aspartic proteases - at positions 25–27/25′–27′. Two important domains can be found near the active site: the so called “fireman’s grip”—a network of hydrogen bonds that supports the core’s rigidity—and the flaps that cover the active site (Figure 3) [32,33]. The flaps are composed of flexible β-hairpins and control access of the substrate or inhibitor to the active site, depending on their conformation, making them an essential element for modulation of enzymatic activity. These can be found in a number of different conformations, including closed, semi-open, and open, which exist in equilibrium but the semi-open state appears to be predominant [34]. It is thought that the flaps open to accommodate the substrate or inhibitor and then close when PR forms a complex with such molecules; several models have been proposed for their opening and closing. For example, it is believed that the curling of the flap tips in the hydrophobic walls of the active site creates enough space for the substrate to fit in, and the active site acquires a negative charge in this conformation that would facilitate interaction with positively charged substrates. Finally, once the substrate is in position, the flaps can extend over it, thus closing the active site and allowing enzymatic cleavage to occur [35]. Another possibility is that this curled intermediate acts as a transitioning conformation between the closed and semi-open states [34]. In another model, interaction with the ligand makes the flap opening more frequent and stable, and when the substrate is in the correct position in the active site, the flaps switch to a closed state until cleavage occurs [36]. 

### 1.2. Sequence Specificity

A consensus cleavage sequence for HIV-1 PR has yet to be defined. Indeed, its target sequences in the viral Gag and Gag-Pol polyproteins differ significantly from one another and are structurally asymmetric (table in Section 2). Studies on such cleavage sites have shown that PR obtains its substrate specificity by recognizing shape and volume rather than a specific amino acid stretch. Therefore, cleavage is not dictated by the sequence itself but is influenced by both substrate dynamics and interactions between PR and the ligand outside of the active site cleft. Moreover, it has been theorized that H_2_O molecules and hydrogen bonds facilitate substrate tridimensional structure recognition by PR, and interestingly, all the substrates derived from the protease cleavage show an extended and asymmetric β-strand conformation while bound to the active site, giving rise to a consensus envelope called the “substrate envelope”. Furthermore, interaction with the ligand occurs both via the active site and the substrate groove (S-groove)—an active pocket present in PR that allows the enzyme to bind up to 24 residues [37]. While there may not be a clear consensus for the cleavage site, evidence shows that HIV-1 PR may have some preference in the cleavage sequence: the two amino acid residues before and after the cleavage site (P2 and P2′) are thought to be more important in recognition than residues beyond this region, and negatively charged, hydrophobic, and β-branched residues are preferred in these positions. In contrast, there is a preference for large, non-β-branched, hydrophobic and aromatic amino acids in sites P1 and P1′. Pro, Gly, and basic residues are mainly ill-favored in all four positions while Arg has never been observed in P2, and the same is true for Pro in P1, that can instead be present in P1′ [38,39,40].

### 1.3. PR Activation and Virion Maturation

PR dimerization is instrumental in modulating the activation of its catalytic activity and dictates the place and timing of nascent virion maturation. Precursor dimers form when the protease is still embedded in the Gag-Pol polyprotein, but they are less stable and therefore poorly active. Although PR precursor monomers can self-interact, it has been postulated that they reach a catalytically active conformation in only 3–5% of cases, subsequently triggering an ordered cascade of events [41]. Due to the poor activity of the precursor dimer, the first proteolytic cuts are intramolecular and result in the release of free enzyme from the polyprotein context. Cleavage occurs at the SP1/NC interface (an internal site within the transframe region) and at the transframe/PR interface (Figure 4a). Next, the enzyme is able to remove the RT domain from its C-terminus, thus releasing itself from the polyprotein and acquiring a mature conformation with increased stability and activity. Finally, fully active PR dimers can start to act intermolecularly on their targets on the Gag polyprotein (Figure 4b) [42]. This activation process is therefore highly concentration dependent and fundamental to avoid the untimely formation of mature PR dimers before the recruitment of a sufficient amount of Gag-Pol to the cell membrane and consequent budding of the immature virion—that up to this stage is composed of approximately 2400 Gag molecules and 120 Gag-Pol molecules [40]. Before the action of the PR, the proto-HIV-1 particle is constituted by a spherical or semi-spherical shell of Gag precursor in which the matrix protein (MA) domain lines the viral envelope, the capsid protein (CA) domain forms protein-protein lattice contacts, and the nucleocapsid protein (NC) domain binds the viral genome. Both the immature and mature capsid are formed by hexameric units; however, these lattices change drastically upon maturation and proteolytic cleavage by the viral PR (Figure 4c). This maturation process is a complex concert of different events: first, cleavage between the spacer peptide 1 (SP1) and NC separates the latter—bound to the viral genome—from the MA-CA lattice, afterwards, MA is separated from the CA lattice, followed by cleavage at the spacer peptide 2 (SP2)/p6 interface that frees the NC from the p6 region. The final cuts are at the N-terminus of the spacer peptides, SP1 and SP2, thereby releasing them from the CA and NC (Figure 4) [40].

## 2. Cellular Targets of HIV-1 Protease

Despite that the role of HIV-1 PR in viral maturation has always been recognized as a crucial event in the virus life cycle, its involvement in the cleavage of cellular factors has been greatly debated. Detection of HIV-1 polyprotein processing on the cytoplasm of HIV-1-infected cells 5 days post-infection suggested possible activity on host cell targets at later stages of infection [43]. However, since an infectious virion brings only 120 copies of PR inside the host cell, the dependency on dimerization for PR activation makes it unlikely to have high protease activity in infected cells before strong de novo Gag-Pol precursor synthesis [40]. Therefore, while it is difficult to hypothesize massive activity on host factors during the early phase of infection (pre-integration), there is growing evidence that HIV-1 PR is involved in modulation of host cell functions at later stages of infection (post-integration). While the lack of a clear consensus cleavage sequence makes it extremely difficult to predict its possible targets inside the cell, a number of studies have applied high-throughput approaches, such as affinity tagging and purification mass spectrometry, to identify possible viral protease targets [38,44]. Some of these targets have been extensively validated in follow-up studies (Table 1), suggesting that HIV-1 PR targets a number of cellular factors to modulate host cell function, thus promoting viral replication and persistence.

In one of the largest HIV-1 interactome studies, Jäger et al. identified 497 viral-host protein interactions from transfected HEK293T and Jurkat cells. Among them, 67 proteins were shown to interact with PR (Figure 5, Appendix A) [44]. Similarly, in a following proteomics-based study, researchers were able to identify more than 140 putative cleavage sites for HIV-1 PR from Jurkat cell lysates incubated in vitro with bacterially expressed PR (Appendix A) [38].

### 2.1. Role of the Protease in Protein Synthesis Modulation

A number of studies have contributed to shine a light on the possible multifaceted role of HIV-1 PR in different steps of its viral life cycle and paved the way to in depth characterization of its cellular targets (Figure 2). The best characterized host-cell process targeted by HIV-1 PR is arguably protein synthesis. During infection, several factors involved in cap-dependent translation are cleaved, resulting in a decrease in cellular cap-dependent translation in favor of late internal ribosome entry site (IRES)-dependent viral protein production [48,50,51,55,56]. This is also supported by a strong reduction in cellular translation observed after exogenous expression of PR in COS-7 cells [48]. Indeed, the HIV-1 RNA genome harbors one IRES within the 5′ LTR [57,58] and one in the *gag* gene (Figure 1) [59], which are especially important for translation of late gene products. Nucleotides 1-270 from the *gag* transcript have been shown to retain cap-dependent translation initiation properties [57]; therefore, since all spliced and unspliced HIV-1 transcripts possess the same 289 nt-long 5’UTR, they could all internally initiate translation. However, the latter has only been observed from the *gag*, *tat*, *vpr*, *vpu*, *vif*, and *nef* leaders, and their activity has been shown to be highly variable [60]. While the translation of viral transcripts is heavily cap-dependent during the first 48 h of the viral life cycle, IRES-mediated translation importantly contributes to viral protein synthesis after that time [55]. Furthermore, the addition of recombinant PR to rabbit reticulocytes for in vitro translation assays caused a 10-fold reduction of cap-dependent protein synthesis in vitro while barely affecting IRES-driven translation and a 4-fold increase in translation of a synthetic mRNA encoding for the HIV-1 5′ leader sequence, *gag* and *pr* [48].

In addition, HIV-1 infection of C8166 cells (human leukemia T cells) resulted in a decrease in total protein synthesis starting from 2 days post-infection. This phenomenon was associated with both a decrease in eukaryotic translation initiation factor 4G (eIF4G) levels and the appearance of eIF4G cleavage products and was completely inhibited by treatment with Saquinavir—thus proving the ability of PR to cleave eIF4G [48]. Based on experiments incubating recombinant PR with HeLa cell extracts, the eIF4G cleavage site was identified between residues 678-9, 681-2, and 1086-7, physically separating the eIF4E and eIF3 binding moieties of eIF4G and functionally impairing its ability to participate in cap-dependent translation (Figure 6) [48]. A further contribution to cap-dependent cellular translation inhibition is the HIV-1 PR-mediated cleavage of poly(A) binding protein (PABP) at positions 237/238 and 477/478 (Figure 6). Such cleavage has been observed both in HIV-1-infected MT-2 cells and in BHK-21 cells transiently expressing PR, taking place in a saquinavir-sensitive manner [50]. Therefore, HIV-1 PR catalytic activity further contributes to host cell cap-dependent translation shut-off, thereby inhibiting poly(A)-dependent initiation of translation by disrupting the synergy between the poly(A) tail and the cap in cellular mRNAs [56].

Additionally, HIV-1 infection also stimulates the cleavage of translation factor eIF3d, which, upon transient expression of PR in HEK 293T cells, is specifically targeted between residues 114 and 115 with an efficiency close to that of Gag-Pol processing [44]. The physiological significance of eIF3d cleavage might extend from the cap-dependent host cell translation shut-off described above, since its knockdown boosted infectivity upon single round infection with VSV-G pseudotyped HIV-1 particles and promoted accumulation of reverse transcription products [44], thus suggesting an antiviral role for eIF3d during the early stages of viral infection, before proviral integration (Figure 2). It has been hypothesized that eIF3d is cleaved upon viral infection by PR present in incoming virions, possibly explaining an earlier study in which 1 h pre-treatment of H9 cells with a HIV-1 PR inhibitor (UK-88947) strongly inhibited HIV-1 DNA synthesis 18 h post-infection, thus suggesting the requirement of HIV-1 PR for optimal viral genome RT processing and integration [61]. However, the importance of HIV-1 PR’s catalytic activity during the early phase of the viral life cycle has been heavily debated, and the role of incoming PR during HIV-1 infection is still questionable [62,63,64,65,66]. Apart from factors directly involved in translation initiation, HIV-1 PR is able to cleave another protein responsible for the regulation of cellular translation: the antiviral kinase general control non-derepressible-2 (GCN2). GCN2 phosphorylates eIF2, hence halting AUG-dependent translation in response to several stimuli—such as amino acid or serum deprivation, UV light irradiation, and viral infection [51]. During the late phase of infection, the abundant production of viral mRNAs can trigger GCN2 activation, thus resulting in the inhibition of both cap- and IRES-dependent protein synthesis, with a detrimental effect on viral replication. Accordingly, infection of PBMC with HIV-1 resulted in a 45% reduction in GCN2 at 4- and 5-days post-infection, which was prevented by the addition of saquinavir. In addition, GCN2 cleavage was observed as early as 4 h post-transfection in BHK-21 cells transfected with an exogenous plasmid encoding for HIV-1 PR [67]. By cleaving GCN2, HIV-1 PR prevents the generalized inhibition of AUG-dependent translation that would negatively impact the viral life cycle (Figure 2). Therefore, HIV-PR is able to target multiple levels of the translation process: on the one hand, it causes a decrease in cap-dependent cellular protein production, thus favoring IRES-mediated translation of unspliced viral transcripts thanks to the cleavage of eIF4G; on the other hand, it ensures abundant viral product translation by preserving AUG-dependent translation thanks to cleavage of GCN2.

### 2.2. PR and Apoptosis

HIV-1 modulates host cell survival at several levels thanks to the action of viral proteins such as Tat, Env, Nef, Vpr, and Vpu [68,69], all of which are endowed with both stimulatory and inhibitory properties. Likewise, there is growing evidence that HIV-1 PR is similarly involved in the modulation of programmed cell death (Figure 7). Given its remarkable effect on translation (see above), it is not surprising that exogenous expression of HIV-1 PR is highly toxic in a number cellular systems [70,71,72]. In addition, HIV-1 PR has been shown to stimulate apoptosis via several mechanisms, such as cleavage of anti-apoptotic factor Bcl-2 [49], procaspase 8 [47], and of a number of mitochondrial proteins [73], thus highlighting its ability to interfere with more than one cell death pathway. The first proof of the involvement of HIV-1 PR in the modulation of apoptosis came from the identification of Bcl-2 amongst its targets. Cleavage was first characterized in vitro and confirmed by western blot analysis of COS-7 cells 24 h after having been transfected to co-express Bcl-2 and HIV-1 PR [49]. Cleavage of Bcl-2 likely promotes apoptosis activation since its overexpression in HIV-1-infected lymphocytes protected cells from apoptosis [49]. In this context, another important target of HIV-1 PR is procaspase 8, the cleavage of which generates the peculiar cas8p41 fragment, which is able to induce apoptosis in a caspase 9- and Bak/Bax-dependent manner when exogenously expressed in cells, similarly to what is observed upon transient expression of HIV-1 PR. However, the cas8p41 fragment is not detectable in all cells exhibiting infection-induced apoptosis, further highlighting that HIV-1 possesses more than one mechanism to induce this kind of programmed cell death [47]. HIV-1 PR also destabilizes mitochondrial membrane integrity. When expressed in HeLa cells, the viral protease localized in the mitochondria and in vitro experiments showed its ability to cleave a number of mitochondrial proteins, such as Tom22, VDAC, and ANT. Furthermore, PR was shown to interact with breast carcinoma-associated protein 3 (BCA3) in the mitochondria [73]. This cellular protein is known to interact with several cellular factors, such as PKAc, Nf-κB, p73, and apoptosis inducing factor (AIF); however, its role in the cell’s physiology remains largely unknown. Despite the fact that interaction between HIV-1 and BCA3 does not result in cleavage, co-expression of the two proteins in HEK 293T cells was associated with increased apoptosis [73]. Interestingly, BCA3 is incorporated in HIV-1 virus like particles (VLPs) [74], which has no effect on viral infectivity and may be mediated by the interaction between the BCA3 C-terminal domain and PKAc that is incorporated into the virion [75]. It has been observed that BCA3 mutants lacking the C-terminal domain are not incorporated into nascent virions, advancing the hypothesis that PKAc acts as a target for this cellular protein [74].

Less clear is the role of HIV-1 PR-mediated cleavage of NF-κB1, a member of the nuclear factor kappa-light-chain-enhancer of activated B cells (NF-κB) family, in apoptosis modulation. Indeed, NF-κB can act both as an anti-apoptotic agent as well as a promoter of cell death [76]. HIV-1 PR specifically cleaves NF-κB1 (also known as p105), the cytoplasmatic precursor of the NF-κB subunit p50; cleavage occurs between residues F412 and P413 and has been observed both in human T-cell extracts treated with HIV-1 PR and in COS cells transiently co-expressing p105 and Pol. In doing so, PR increases the levels of readily available NF-κB and facilitates its translocation into the nucleus [52]. At the moment, in the absence of additional experimental evidence, it is extremely difficult to speculate the exact outcome of this particular cleavage due to the complexity of NF-kB regulation, but it very likely represents an additional way through which HIV-1 exerts its host-modulating activity via PR.

As alluded to above, HIV-1 PR is also capable of suppressing programmed cell death pathways, such as by targeting the nuclear Dbf2-related kinases NDR1 and NDR2 [46,53]. Their cleavage was observed in HEK293T cells transfected with HIV-1_NL4-3_ proviral DNA, in which expression of PR resulted in processing of almost 50% of endogenous NDR1. These kinases have been associated with different cellular mechanisms, such as morphological changes, the cell cycle, apoptosis, and innate immunity [77]. For example, NDR1/2 are activated by the death receptor and their knockdown reduced cell death and apoptosis [68]. Interestingly, NDR1 negatively affects the TLR9-mediated response and a lack of this kinase has been associated with enhanced proinflammatory cytokine production induced by CpG in vivo. Likewise, NDR2 also seems to influence the TLR9 response and reduced activity of NDR2 was correlated with stronger secretion of IL-6 mediated by CpG [77]. Furthermore NDR1/2 act by promoting the ubiquitination of MEKK2 via Smurf1, thus stimulating the production of TNF-α, IL17, and IL6 and the activation of NF-κB. Moreover, mice lacking NDR1 and infected with *E. coli* produced higher levels of such cytokines and had a higher mortality rate [78,79]. NDR1 is also able to promote the production of STAT1 inside cells, hence enhancing interferon-mediated immunity. Additionally, knockdown/overexpression of NDR1 showed a direct involvement in antiviral defense, since NDR1-deficient macrophages from mice showed decreased production of proinflammatory cytokines and ISG expression, while the opposite was observed when NDR1 was overexpressed in murine RAW264.7 cells [80]. Cleavage of NDR1 and NDR2 in their hydrophobic domains ablated their transphosphorylating activity but did not affect their autophosphorylation; furthermore, it altered the cellular distribution of the truncated portion of NDR2 that was shown to abnormally migrate into the nucleus. Additionally, these enzymes were shown to be incorporated inside the virion along with other cellular kinases, such as ERK2/MAPK and C-PKA, which are reportedly involved in infectivity modulation [46]. The role of NDR1 and NDR2 inside the cell is complex and still not completely characterized, but their involvement in apoptosis induction and innate immune responses suggests that their cleavage by HIV-1 PR and the consequent alteration of their activity serves as another layer of immune evasion employed by the virus.

Similarly, HIV-1 PR is also able to cleave the receptor interacting kinases RIPK1 and RIPK2 [45,53], which are potentially implicated in cell survival and the innate response. Indeed, RIPK1 is involved in apoptosis and necroptosis activation, while RIPK2 is involved in the activation of MAVS. However, although both kinases were efficiently cleaved in vitro, only endogenous RIPK1 was cleaved in a Jurkat cell line upon doxycycline-mediated induction of HIV-1 PR and in HEK293T cells infected with a VSVg pseudotyped HIV-1 NL4.3 and Sup-T1 cells infected with replication-competent HIV-1 NL4.3 at 24 and 48 h post-infection, respectively. In all three cases, treatment with a PR inhibitor prevented cleavage of the cellular protein; moreover, inhibition of either reverse transcription or integration completely abrogated RIPK1 cleavage after infection of Sup-T1 cells with HIV-1 NL4.3, proving that RIPK1 was mainly cleaved by post-integration expressed PR rather than by PR present in incoming virions. RIPK1 cleavage was placed between residues 462–463 thanks to mass spectrometry experiments, thereby separating its kinase and death domains. Therefore, despite the fact that overall levels of endogenous RIPK1 were not reduced, the emergence of such defective RIPK1 forms likely promoted cell survival of HIV-1-infected cell [45].

### 2.3. Effect of PR on Innate Defenses

Viral and bacterial infections are controlled and cleared by the immune system, which comprises two fundamental branches: the innate and adaptive immune systems. The innate immune system is the first line of defense against pathogens and its response is very rapid but lacks immunological memory. On the other hand, the adaptive immune system requires several days to mount a response but maintains a memory for the antigen, thus enabling a more rapid and specific response when rechallenged.

Synergy between the innate and adaptive immune responses is key to efficient clearance of invading pathogens. Pathogen-recognition receptors (PRRs) present in infected cells have the ability to recognize specific viral pathogen-associated molecular patterns and initiate a signaling cascade that leads to the production of type I and type III interferons (IFN) as well as proinflammatory cytokines. These soluble molecules induce both cell-intrinsic innate immune mechanisms as well as the recruitment and activation of other types of immune cells, thus achieving restriction of viral replication and spread. HIV infection is detected by several PRR in macrophages and dendritic cells, including Toll-like receptor 7 (TLR7), interferon inducible protein 16 (IFI16), and GMP-AMP synthase (cGAS), which have been shown to interact with other sensors such as polyglutamine binding protein 1 (PQBP1) and NONO [81,82,83,84,85]. Both IFI16 and cGAS engage STING, which in turn recruits signaling cofactors such as TANK-binding kinase 1 (TBK1) and IKK-α/β to activate IRF3 and NF-κB, thereby inducing the gene expression of type I IFN and proinflammatory cytokines [82,86,87].

Notably, TBK1 was shown to be cleaved by HIV-1 PR, first via a cell-free alpha screen assay aimed at identifying potential new kinase targets of the viral PR, and subsequently in HEK 293T cells transiently expressing Gag-Pol and TBK1 [53]. Cleavage between residues L683 and V684 prevents its phosphorylation, which is fundamental for its activation, thus suppressing its ability to activate IRF3. Interestingly, treatment of T7 cells harboring latent HIV-1 infection with Bryostatin-1—an agent capable of strongly reactivating HIV-1 from latency—and the PR inhibitor indinavir resulted in increased IFNβ1 transcription and upregulation of ISG-15 mRNA compared to untreated cells, suggesting a possible role of HIV-1 PR as a negative regulator of IFN-1 and the innate immune response [53].

Finally, as seen with the interaction with BCA3, cleavage is not the only mechanism by which HIV-1 PR is able to exert its host-modulating action. Indeed, PR is able to downregulate the protein level of RIG-I in a proteolysis-independent manner [88]. RIG-I is a pattern recognition receptor (PRR) involved in the recognition of viral ssRNAs, which cause the subsequent activation of antiviral defenses. Interaction between RIG-I and HIV-1 PR causes the relocalization of the former to the insoluble membrane fraction and consequent lysosomal elimination [88], thus contributing to the complex interplay between HIV-1 and its host.

### 2.4. Cleavage of Virion-Incorporated Restriction Factors

Remarkably, HIV-1 PR targets two restriction factors that are incorporated into the virion to interfere with virus fitness (Figure 2): YTHDF3 and APOBEC3H haplotype II splice variant 200 [89,90,91]. The N^6^-methyladenosine (m^6^A) reader protein YTHDF3 is incorporated into the nascent virion via NC interaction and is subsequently cleaved by PR [89]. YTHDF3 recognizes RNA molecules bearing the m^6^A modification and can promote either their translation or degradation. HIV-1 infection upregulates m^6^A modification, which appearance in viral RNAs is linked to enhanced nuclear export and increased viral protein production. NC-mediated incorporation of YTHDF3 was observed by production and analysis of virus like particles generated by co-transfection of Gag and FLAG-YTHDF3 constructs in HEK293T cells. Cleavage was investigated in virions produced in A3R5-Rev-GFP T cells infected with the NL4-3 HIV-1 strain and harvested 12 days post-infection. Analysis of both viral and cell lysates showed that the former presented a majority of shorter processed YTHDF3 compared to the full-length version that was instead detected in cell lysates. Interestingly, YTHDF3 is believed to act as a restriction factor for HIV-1 since its knockout in A3R5-Rev-GFP cells and CD4+ T cells resulted in higher susceptibility to infection, whereas its overexpression negatively impacted viral infectivity [89]. Therefore, its cleavage by HIV-1 PR plays a role in incrementing virus fitness.

APOBEC3H haplotype II splice variant 200 (A3H-II SV200) is similarly cleaved by HIV-1 PR in the nascent virion. Other members of the APOBEC3 family, such as APOBEC3D, APOBEC3F, and APOBEC3G, have a strong anti-HIV-1 effect and are incorporated into the virion where they hypermutate and inactivate the viral genome, thus catalyzing the deamination of C to U [17]. Their antiviral activity is usually counteracted by HIV-1 accessory protein Vif, which recruits the host E3 ubiquitin ligase complex on APOBEC3 proteins, thus promoting their degradation [17]. In addition, A3H-II SV200—one of the APOBEC3 family members most susceptible to genetic variability in the population—is cleaved by PR. In particular, haplotype II is one of the few functional variants of A3H and can be found in 82% of HIV-1 pandemic area populations in sub-Saharan Africa; additional variability of this protein comes from possible splice variants of this allele, among which SV200 is the most active in restricting HIV-1 infectivity. Cleavage of A3H-II SV200 was detected in Vif-deficient virions harvested from infected HEK293T cells transfected with a A3H-II SV200 construct, and was confined inside the viral particle; PR-mediated cleavage of this factor likely involves its C-terminal region and decreases its activity [90]. Therefore, despite the fact that HIV-1 is able to potently counteract the action of APOBEC3 proteins by the activity of Vif, it appears that A3H-II SV200 is further specifically targeted by HIV-1 PR.

Lastly, HIV-1 PR was shown to target Lyric (also named metadherin or AEG-1), a protein involved in several cellular pathways, such as NF-κB, Ras, Wnt, PI3K, tumor growth and metastatization. Lyric directly interacts with Gag and is incorporated in the nascent virion where it is cleaved by the PR. Experiments with infected MT-4 cells showed a 6-fold enrichment of lyric in virions compared to cell lysates. Furthermore, virion-incorporated lyric lacked the C-terminal portion, while the full-length protein was recovered in PR-deficient virions and cells [91]. Although the meaning of this interaction is still unknown, it has been reported that expression of lyric was correlated with increased Gag expression and viral infectivity, and it possibly plays a role in HIV-associated neurocognitive disorder (HAND) [92].

### 2.5. PR-Mediated Cytoskeleton Modulation

Nearly every virus has evolved mechanisms to interact with the cytoskeleton to some extent, for example, exploiting it for viral replication or heavily altering its properties upon viral-induced cell transformation [93]. Likewise, HIV-1 interacts with the cytoskeleton in several different ways, one of which is via PR-mediated cleavage of cytoskeletal proteins. Indeed, the viral protease was demonstrated to cleave several cytoskeletal proteins: for example, vimentin was cleaved after PR microinjection in human skin fibroblasts with subsequent alteration of nuclear morphology and chromatic condensation [94]. Actin was also cleaved in vivo in A3.01 T lymphocytes infected with HIV-1 (LAV-1_BRU_), but the protease was only able to cleave circulating globular actin and not the cytoskeletal fraction [95,96]. Intriguingly, HIV-1 greatly depends on actin in early steps of the replication cycle, from attachment and fusion of the virion to the cellular membrane to nuclear localization of the pre-integration complex [97]. Notably, disruption of the actin cytoskeleton by cytochalasin D inhibits viral entry and reverse transcription. Furthermore, actin has also a role in later stages of the infection, where actin-depolymerizing agents were shown to inhibit HIV-1 assembly [93]. However, even though many aspects of the close-knit relationship between HIV-1 infection and the cytoskeleton are well characterized and involve many viral proteins [98], the functional relevance of the cleavage of cytoskeletal proteins by PR is still unclear. It may serve as a way for the virus to better navigate the host cell or yet another way the virus has evolved to interfere with more complex regulatory pathways.

### 2.6. Cleavage of Host Factors by HIV-2 PR

HIV is categorized into two distinct subtypes, HIV-1 and HIV-2, both of which share similar transmission routes and are able to cause AIDS. However, the two viruses present important differences in terms of epidemiology, diagnosis, and management [99]. Indeed, HIV-2 is characterized by lower transmissibility and reduced likelihood of progression to AIDS and has mainly remained confined to West-African countries. Despite possessing a similar three-dimensional structure, HIV-1 and HIV-2 proteases share rather low amino acid sequence identity, between 38% and 49% depending on the viral strains [100]. Accordingly, the two enzymes differ in both sensitivity to inhibitors and gag-pol precursor sequence specificity, especially at the P2 positions of peptide substrates [101]. Therefore, it cannot be taken for granted that host factors targeted by HIV-1 PR are similarly cleaved by HIV-2 PR. So far, very little data are available on the topic and only three cellular targets have also been tested for cleavage by HIV-2 PR. Intriguingly, all three are factors involved in cellular translation and are cleaved by both proteases: GCN2, eIF4G, and PABP [50,51,102]. However, while incubation of GCN2 with HIV-2 PR generates fragments identical to those obtained after incubation with HIV-1 PR [51], the number and exact position of cleavage sites on both eIF4G and PABP differ slightly between the PRs from the two viruses [50,102]. These findings suggest that although HIV-1 and -2 most likely target the same cellular pathways via their PRs, the molecular details are likely divergent.

### 2.7. HIV-1 PR Inhibitors

Detailed structural knowledge of HIV-1 PR and its substrates led to the development of specific protease inhibitors (PIs) [103]. To date, nine different PIs have been approved for clinical use: saquinavir (SQV), ritonavir (RTV), indinavir (IDV), nelfinavir (NFV), fosamprenavir (APV), lopinavir (LPV), atazanavir (ATV), tipranavir (TPV), and darunavir (DRV). All PIs beside tipranavir are competitive peptidomimetic inhibitors that mimic the natural substrate of the viral PR and cannot be cleaved by HIV-1 PR [104]. PIs have been designed to bind to the substrate-binding region of the mature viral PR dimer with high affinity. Despite the fact that PIs have been widely used in anti-HIV-1 HAART in previous decades, their use has been recently reduced due to issues related to toxicity, selection of viral resistant strains, and approval of a number of drugs acting on alternative targets. Indeed, more than 25 different medications from six different classes are available for treatment of HIV-1-infected patients. Nowadays, the standard of care for most treatment-naïve patients is composed of two nucleoside RT inhibitors, such as tenofovir and emtricitabine, in combination with one non-nucleoside RT inhibitor or with an IN inhibitor. Furthermore, a number of first-generation PIs, such as SAQ, APV, and IDV, are no longer used for several reasons, including a low genetic barrier and concomitant selection of resistant viruses, severe side-effects, as well as inefficacy as compared to more recently approved compounds. However, PIs are still widely used in salvage therapy for patients who fail initial HAART [105], and boosted PI in combination with an optimized nucleoside RT inhibitor backbone is recommended as a preferred second-line regimen for people living with HIV for whom Dolutegravir-based regimens are failing [106].

### 2.8. HIV-1 PI Resistance Mutations

Although second-generation PIs, such as APV, LPV, ATV, TPV, and DRV, have an intrinsic higher genetic barrier to the development of resistance, PI-resistant mutants occasionally arise, including multi-PI-resistant strains [107]. Most PI primary resistance mutations map to PR itself, and in particular, cluster where PIs protrude beyond the substrate-binding pocket, involving PR residues that interact with the inhibitor but not with the natural substrate. This is possible since PIs interact with a larger PR surface as compared to the gag polyprotein cleavage sites. The most common primary mutations are D30N, G48V, I50L/V, V82A/F/L/S/T, I84V, and L90M (see Table 2). However, such viral mutants are also impaired in viral replication because the PI-resistant PR has reduced affinity for the gag polyprotein, which is therefore inefficiently processed. Indeed, resistance to PIs is a compromise between resistance and PR enzyme function. Hence, PR primary mutations are often followed by secondary mutations (also known as compensatory mutations), which can restore viral replication and/or enhance drug resistance [108]. Such mutations can be found in the viral PR itself as well as in the Gag substrate [109]. Intriguingly, a few Gag substrate mutations are able to confer PI resistance in the absence of additional PR mutations and are therefore primary drug resistance mutations, acting by increasing the affinity of the PR-gag interaction [110]. However, PI-resistant Gag compensatory mutations often occur in PR cleavage sites (CS), thus greatly enhancing processing by the PI-resistant mutant PR and restoring viral fitness to a certain extent [109,111,112,113,114]. The most common Gag CS compensatory mutations occur at the NC/SP2 CS (A431V and I437V), which is associated with the PR primary mutation V82A, and the SP2/p6 CS (L449F and P453L), which is associated with PR primary mutations I50V and I84V [111,115,116] (see also Figure 4b and Table 2). Although it has never been tested experimentally, it is reasonable to speculate the PI-resistant PR mutants selected during PI treatment may be impaired in their ability of cleave host factors, regardless of Gag CS compensatory mutations, thus potentially explaining why the latter increase Gag polyprotein processing but do not fully restore viral fitness [117]. Indeed, human genes coding for such targets are not capable of rapidly mutating and actively replicating HIV-1 infected cells are ultimately destined to cell death.

### 2.9. Implication of HIV-1 PR Mediated Host Factor Cleavage for Antiviral Therapy

A relatively new class of experimental HIV-1 drugs, so-called maturation inhibitors (MI), act by preventing the cleavage of a specific Gag junction by HIV-1 PR [118]. Maturation inhibitors do not directly inhibit HIV-1 PR but block cleavage of the Gag polyprotein by directly recognizing the CS between CA and SP1, thus preventing its processing by PR. Therefore, maturation inhibitors cause accumulation of the CA-SP1 precursor, ultimately impairing viral replication. However, Bevirimat (BVM), the first MI developed, failed phase IIb clinical trials due to the rapid emergence of Gag mutations at the CA-SP1 junction. The clinical development of GSK3532795, a second-generation derivative of BMV with a higher genetic barrier, was terminated due to high toxicity. In this context, it would be tempting to hypothesize that compounds similarly able to prevent host factor cleavage could represent a target for therapeutic development. Indeed, unlike BVM and other maturation inhibitors, such drugs should not allow rapid emergence of resistant mutants.

## 3. Conclusions

HIV-1 PR activity was long thought to be restricted only to immature virions during or after budding in order to catalyze maturation of nascent viral particles, but as we reported in this review, that is just a small part of the whole picture. This viral enzyme has been proven to interfere with several host physiological processes with the aim of facilitating the progression of viral infection, and although there is a growing body of evidence on this matter, the depth of what is unknown is still baffling. Open questions remain in relation to the functional role of several reported PR targets as well as their exact cleavage location and timing in infected cells. Moreover, these studies are far from perfect and present some limitations, such as the use of cell lines that are not naturally targeted by the virus or employment of laboratory-adapted viral strains of HIV-1. Additionally, transient transfection does not depict an accurate model of HIV-1 infection. Studies on primary cell lines are required to confirm some of these results and understand their effect on CD4+ lymphocytes and monocytes. In light of these facts, it is clear that there is still a huge gap in knowledge in understanding how HIV-1 infection affects a host cell, and in this regard, PR activity may have been grossly underestimated and needs to be thoroughly researched.

## Figures and Tables

**Figure 1 viruses-15-00712-f001:**
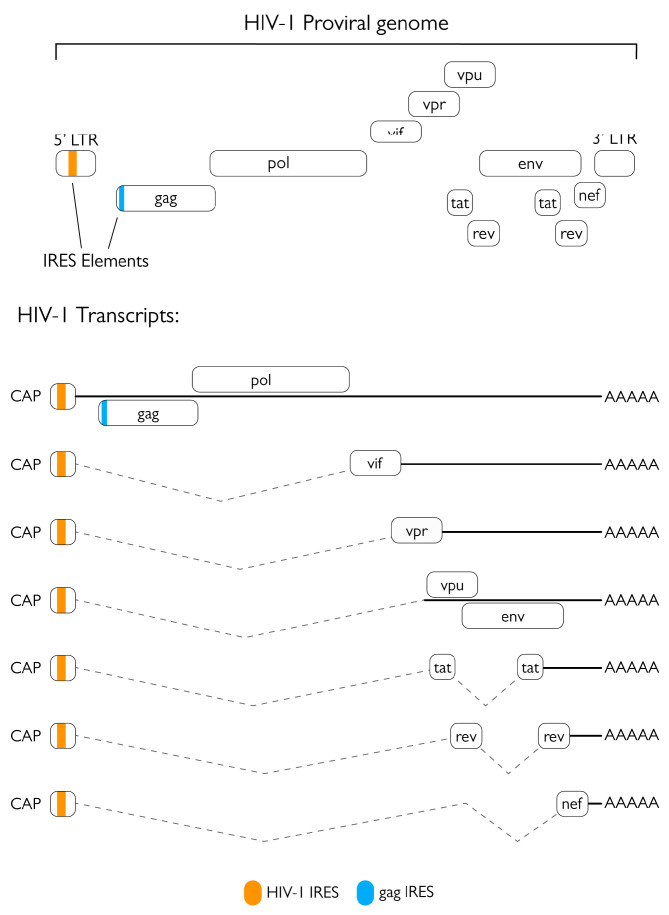
Schematic representation of IRES locations in the HIV-1 genome and transcripts.

**Figure 2 viruses-15-00712-f002:**
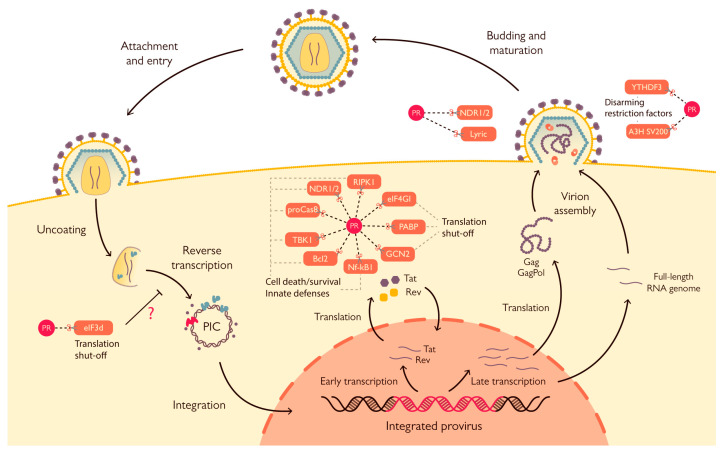
HIV-1 replication cycle with targets cleaved by HIV-1 protease. The exact timing and location of the activity of HIV-1 PR are still not completely characterized. It is speculated that due to its possible anti-RT activity, *eIF3D* may be an early target of PR, and inactivating this factor may serve as a way to escape antiviral defenses of the cell and hinder cap-dependent translation. The other intracellular targets are most likely cleaved by post-integration synthetized PR, among these targets there are several proteins involved in protein synthesis that are responsible for either cap-dependent translation initiation (*eIF4G*, *PABP*) or translation regulation (*GCN2*). The other major protein cluster that is targeted by the viral enzyme represents proteins involved in cell death and the innate defenses of the cell (*Bcl2*, *Procaspase8*, *NF-κB1*, *NDR1/2*, *RIPK1*, *TBK1*). Lastly, HIV-1 PR was shown to cleave host proteins that are incorporated into the nascent virion, two of which exert antiviral activity (*A3H SV200*, *YTHDF3*) while the specific function of the other two (*NDR1/2*, *Lyric*) remain unknown.

**Figure 3 viruses-15-00712-f003:**
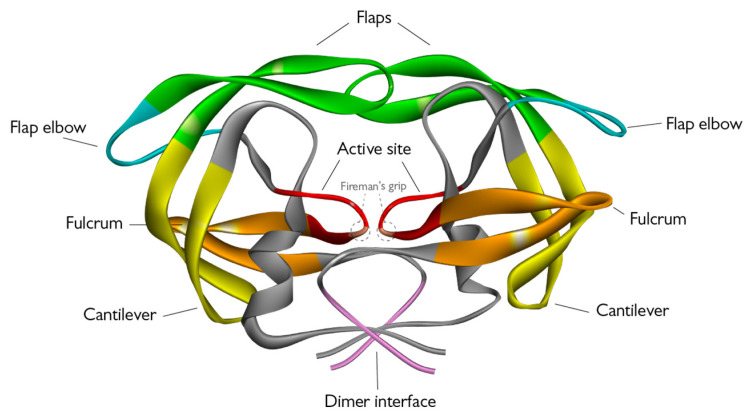
3D structure of HIV-1 PR (PDB ID 6O48) visualized as a ribbon diagram with functional domains highlighted in different colors [34]: protease flaps in green, flap elbows in turquoise, fulcrum in orange, cantilever in yellow, dimer interface in pale magenta, and active site in red.

**Figure 4 viruses-15-00712-f004:**
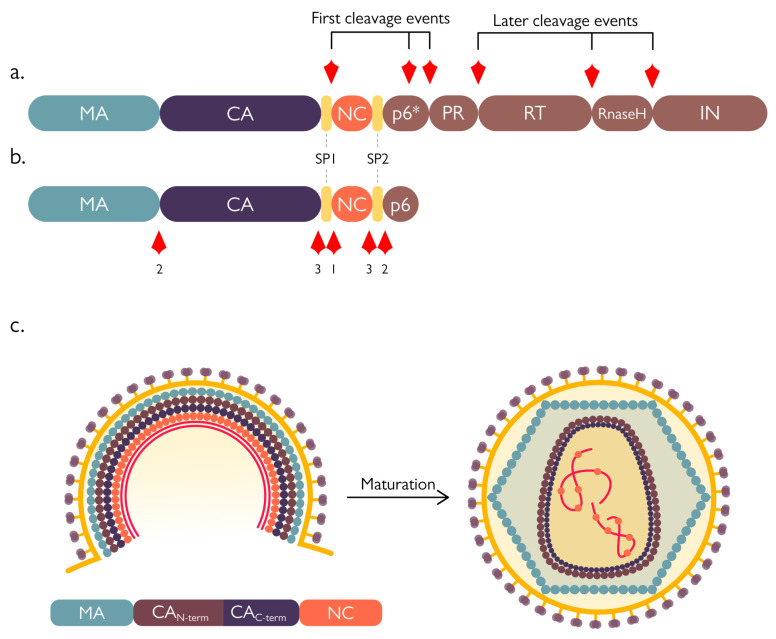
Processing of HIV-1 polyproteins by the viral protease. (**a**) Ordered processing of Gag-Pol by the poorly-active protease (PR) precursor still embedded in the polyprotein context. The first cleavage events aim at separating PR from the polyprotein at its N-terminus (separating the PR from the transframe domain p6*), later events finally free the protease from the rest of the precursor. (**b**) Processing of Gag by the mature PR. Maturation occurs in a precise and ordered manner (events order is numbered 1–3), with the first cleavage occurring between the nucleocapsid (NC) and spacer peptide (Sp)1, then between the matrix (MA)/capsid (CA) and SP2/p6, and lastly, the CA and NC are separated from the spacer peptides. (**c**) Schematic representation of virion maturation.

**Figure 5 viruses-15-00712-f005:**
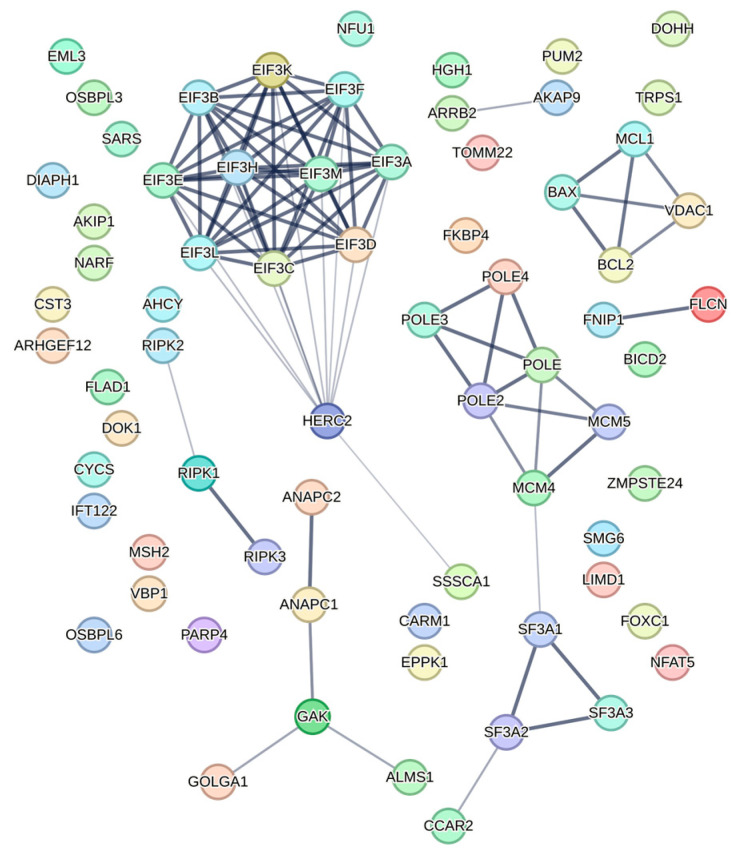
Graph of identified HIV-1 PR interactors from Jäger et al. represented by STRING [54].

**Figure 6 viruses-15-00712-f006:**
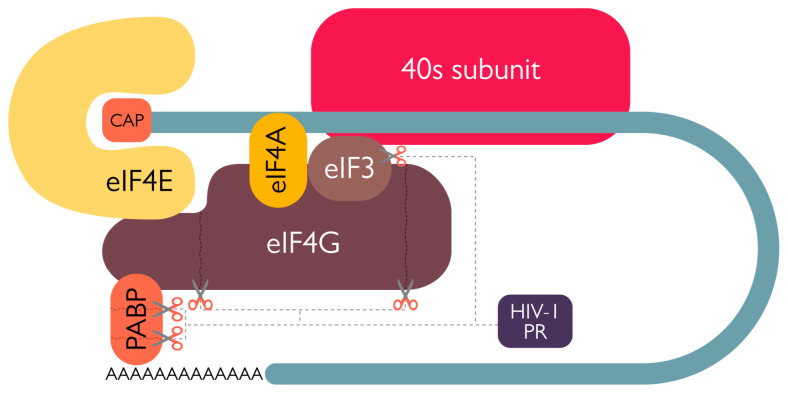
Schematic representation of eukaryotic 43S pre-initiation complex with HIV-1 PR cleavage targets.

**Figure 7 viruses-15-00712-f007:**
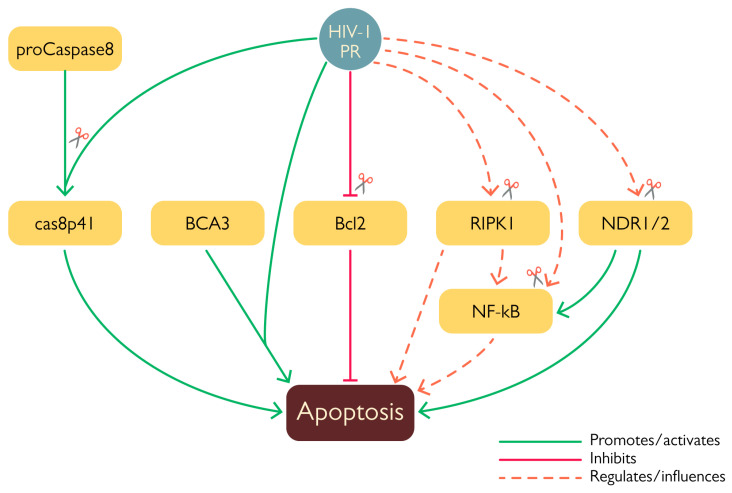
HIV-1 PR modulation of apoptosis via cleavage or interaction with cellular factors. The viral protease is able to cleave several proteins involved in apoptosis regulation. By cleaving procaspase 8 into its cas8p41 fragment, the viral enzyme is able to promote programmed cell death. Likewise, cleavage and consequent inhibition of Bcl-2 also contributes to apoptosis activation. Interestingly, the non-proteolytic interaction between PR and BCA3 was also shown to increase apoptosis. Furthermore, HIV-1 PR cleaves several other proteins involved in the activation and regulation of apoptosis. In these cases, the consequences of their cleavage are not as clear but they fit into the complex scheme of HIV-1-dependent regulation of programmed cell death.

**Table 1 viruses-15-00712-t001:** HIV-1 PR cleavage sites validated in cell culture systems. Gag and Gag-Pol domains are expressed as abbreviations as follows: matrix protein, MA; capsid protein, CA; spacer peptide 1, SP1; nucleocapsid protein, NC; spacer peptide 2, SP2; transframe region, TF; protease, PR; reverse transcriptase, RT; RNAse H domain of the reverse transcriptase, RH; integrase, IN.

Protein	Cleavage Sequence
HIV-1 cleavage sites	
Gag (MA-CA)	SQNY^PIVQ [39]
Gag (CA-SP1)	ARVL^AEAM [39]
Gag (SP1-NC)	ATIM^MQRG [39]
Gag (NC-SP2)	RQAN^FLGK [39]
Gag (SP2-p6)	PGNF^LQSR [39]
Gag-Pol (TF-PR)	SFNF^PQIT [39]
Gag-Pol (PR-RT)	TLNF^PISP [39]
Gag-Pol (RT-RH)	AETF^YVDG [39]
Gag-Pol (RH-IN)	RKIL^FLDG [39]
Host protein cleavage sites	
RIPK1	PQVL^YQNN [45]
NDR1	KDWV^FINY [46]
NDR2	KDWV^FlNY [46]
proCaspase8	PKVF^FIQA [47]
eIF4GI	KIIA^TVLM [48]
	ATVL^MTED [48]
	RFSA^LQQA [48]
eIF3d	RRNM^LQFN [44]
Bcl2	RRDF^AEMS [49]
PABP	GFVS^FERH [50]
	PRVM^STQR [50]
GCN2	GQDY^VETV [51]
NF-κB1	HYGF^PTYG [52]
TBK1	SNTL^VEMT [53]

**Table 2 viruses-15-00712-t002:** List of FDA-approved HIV-1 protease inhibitors. Mutations are indicated using the single letter code for amino acids.

Protease Inhibitor	FDA Approval (Year)	Notable Resistance Mutations
Saquinavir	1995	G48V, L90M
Ritonavir	1996	Used as pharmacokinetic enhancer
Indinavir	1996	M46I/L, V82A/F/T, I84V
Nelfinavir	1997	D30N, L90M
Fosamprenavir	1999	I50V, I84V
Lopinavir	2000	V32I, I47V/A, L76V, V82A/F/T/S
Atazanavir	2003	I50L, I84V, N88S
Tipranavir	2005	I47V, Q58E, T74P, V82L/T, N83, I84V
Darunavir	2006	I47V, I50V, I54M/L, V76V, I84V

## Data Availability

Not applicable.

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
