# Peer review of "Cellular Targets of HIV-1 Protease: Just the Tip of the Iceberg?"

_viruses, 2023, doi:10.3390/v15030712_

Round 1
Reviewer 1 Report
In this article, Centazzo et al. review the impact of HIV-1 protease activity on host cell targets. As they posit, it was earlier believed that the HIV protease action is restricted to the virions, but recent literature suggests that it also has a role in cleaving host proteins. The article is well written with a comprehensive description and discussion around critical aspects of HIV proteases, including their structure, activity, targets, and functions. However, there are a few more gaps that the authors need to fill in to make the article complete:
Majors
1. The authors have comprehensively spoken about the roles of HIV-1 protease. However, with therapeutic interests in mind, the authors need to add a section on Protease inhibitors. In that section, they should also include a subsection or a conclusion to discuss the impact of these findings, which suggest the role of proteases in cleaving host proteins on future therapeutics.
2. The authors have added a brief sub-section on the role of protease in innate defense (Section 2.3). This section can be elaborated by including a more detailed description of the roles in Macrophages and Dendritic Cells as well as in the adaptive immune T cells.
3. For a better background, the authors need to add a brief overview of other components of HIV and their roles as well: Gag, pol, tat, rev, etc., in the introductory section.
Minor:
1. Figure 2: Define the abbreviations of MA, CA, NC, PR, etc., in the figure legends as well.
2. Figure 2: Label Sp1 and sp2.
3. References are not available for many instances. The following error is frequently seen:
“Error! Reference source not found.”.
4. Figure 5: Font style needs to be increased.
5. Figure 6: eIF4G 40 S subunit, eif3 should be in a white (or bright) font to make it legible.
6. None of the figures and tables are cited in the text.
7. The English language is acceptable, but several punctuation errors exist. Professional proofreading is recommended.
Author Response
REVIEWER #1
In this article, Centazzo et al. review the impact of HIV-1 protease activity on host cell targets. As they posit, it was earlier believed that the HIV protease action is restricted to the virions, but recent literature suggests that it also has a role in cleaving host proteins. The article is well written with a comprehensive description and discussion around critical aspects of HIV proteases, including their structure, activity, targets, and functions. However, there are a few more gaps that the authors need to fill in to make the article complete:
Majors
1. The authors have comprehensively spoken about the roles of HIV-1 protease. However, with therapeutic interests in mind, the authors need to add a section on Protease inhibitors. In that section, they should also include a subsection or a conclusion to discuss the impact of these findings, which suggest the role of proteases in cleaving host proteins on future therapeutics.
A. We thank the Reviewer for pointing out this issue in our review. We have added now three sections “HIV-1 PR inhibitors“, “HIV-1 PI resistance mutations“ and “Implication of HIV-1 PR mediated host factor cleavage for antiviral therapy”, to summarize and discuss the themes suggested by the Reviewer.
2. The authors have added a brief sub-section on the role of protease in innate defense (Section 2.3). This section can be elaborated by including a more detailed description of the roles in Macrophages and Dendritic Cells as well as in the adaptive immune T cells.
A. We thank the Reviewer for his/her suggestion, we have added the requested information in the section entitled “Effect of PR on innate defenses”
3. For a better background, the authors need to add a brief overview of other components of HIV and their roles as well: Gag, pol, tat, rev, etc., in the introductory section. A. We thank the Reviewer for such useful suggestion, which has been taken on board by adding a brief overview of HIV-1 proteins and their function to the Introduction section
Minor:
4 Figure 2: Define the abbreviations of MA, CA, NC, PR, etc., in the figure legends as well.
A. We thank the Reviewer for noticing this issue. We have now added abbreviations to the Figure Legend, that now reads “Figure 2. Processing of HIV-1 polyproteins by the viral protease. (a) Ordered processing of Gag-Pol by the poorly-active protease (PR) precursor still embedded in the polyprotein context. The first cleavage events aim at separating the PR from the polyprotein at its N-term, later events finally free the protease from the rest of the precursor. (b) Processing of Gag by the mature PR. Maturation happens in a precise and ordered fashion, with the first cleavage happening between nucleocapsid (NC) and spacer peptide (Sp)1, then matrix (MA)/capsid (CA) and sp2/p6, and lastly CA and NC are separated from the spacer peptides. (c) Schematic representation of virion maturation.”
5. Figure 2: Label Sp1 and sp2.
Figure 2 has been modified accordingly.
6. References are not available for many instances. The following error is frequently seen: “Error! Reference source not found.”
A. We apologize for this issue, which was erroneously generated online during the word to pdf conversion. All references are now correctly formatted in the revised pdf file.
7. Figure 5: Font style needs to be increased.
A. Font size has been increased as requested.
8. Figure 6: eIF4G 40 S subunit, eif3 should be in a white (or bright) font to make it legible.
A. We thank the reviewer for his/her suggestion. Font color relative to both eIF4G and eIF3 has been changed as requested.
9. None of the figures and tables are cited in the text.
A. We apologize for this issue, which was erroneously generated online during the word to pdf conversion (see above). All Figures and Tables are now correctly referenced to in the revised pdf file.
10. The English language is acceptable, but several punctuation errors exist. Professional proofreading is recommended.
A. The manuscript has now been extensively proofread.

Reviewer 2 Report
The review article entitled "Cellular targets of the HIV-1 Protease: just the tip of the iceberg?" by Centazzo et al., comprehends the host targets of HIV-1 PR. The manuscript contains extensive literature with vital information and is well-organized. I have a few comments and suggestions listed below, which will help to improve the quality of the manuscript further.
Overall, there are a few typographical errors that need to be fixed. For example, Line #54 and Line #60 have Error messages.
Authors should include a table of all the host factors with which PR interacts.
It will be nice to have a paragraph/section discussing the major resistant mutations reported in PR.
The authors should also briefly describe if there are any differences reported in HIV-1 and HIV-2 PR targets.
The abstract is lengthy. Authors should reduce the length with vital information on what this review covers.
Authors can consider adding a few lines about the role of PR in cell-to-cell transmission of HIV-1 in section 2.4 or can add a separate section.
Author Response
The review article entitled "Cellular targets of the HIV-1 Protease: just the tip of the iceberg?" by Centazzo et al., comprehends the host targets of HIV-1 PR. The manuscript contains extensive literature with vital information and is well-organized. I have a few comments and suggestions listed below, which will help to improve the quality of the manuscript further.
- We are happy that the Reivewer found our Literature Review of interest, and are grateful for the insightful comments.
- Overall, there are a few typographical errors that need to be fixed. For example, Line #54 and Line #60 have Error messages.
- We apologize for this issue, which was erroneously generated online during the word to pdf conversion. All references are now correctly formatted in the revised pdf file. The manuscript has also been extensively proofread.
- Authors should include a table of all the host factors with which PR interacts.
- We thank the Reviewer for his/her suggestion. Although both host factors which have been shown to interact with HIV-1 PR identified in the Jager study and those which have been validated during viral infection as HIV-1 PR targets were already shown in Figure 3 and Table I, respectively we have now decided to include the full list of HIV-1 interactors also a Table (Supplementary Table I), along with a list of all identified putative cleavage sites identified through positional proteomics by Impens et. al. (Supplementary Figure II).
- It will be nice to have a paragraph/section discussing the major resistant mutations reported in PR.
- We thank the Reviewer for his/her suggestions, the major PI resistant mutations are discussed in a dedicated section “HIV-1 PI resistance mutations”
- The authors should also briefly describe if there are any differences reported in HIV-1 and HIV-2 PR targets.
We thank the Reviewer for raising this point. Although our work is mainly focused on HIV-1 PR, we agree it’s a good idea to also deal with HIV-2. Indeed, the two enzymes share only limited amino acid similarity and this reflects in slight differences in substrate specificity. Unfortunately, being mainly confined to Wester African countries, HIV-2 is far less characterized than HIV-1. Relevant literature on the topic is discussed in the newly-written dedicated section entitled “Cleavage of host factors by HIV-2 PR”.
- The abstract is lengthy. Authors should reduce the length with vital information on what this review covers.
- We thank the reviewer for his/her observation. We have now reduced abstract’s length from 331 to 246 Words. We believe it is much more readable and concise now.
- Authors can consider adding a few lines about the role of PR in cell-to-cell transmission of HIV-1 in section 2.4 or can add a separate section
- We thank the reviewer for this suggestion, we have modified text in the introduction section accordingly.

Round 2
Reviewer 1 Report
The authors have added all my concerns satisfactorily.
Reviewer 2 Report
The authors have made significant changes and the manuscript is in a better shape.